# Mechanical Performance of Artificial Hip Stems Manufactured by Hot Forging and Selective Laser Melting Using Biocompatible Ti-15Zr-4Nb Alloy

**DOI:** 10.3390/ma14040732

**Published:** 2021-02-04

**Authors:** Yoshimitsu Okazaki, Jun Mori

**Affiliations:** 1Department of Life Science and Biotechnology, National Institute of Advanced Industrial Science and Technology, 1–1 Higashi 1–Chome, Tsukuba, Ibaraki 305-8566, Japan; 2Instron Japan Company Limited, 1-8-9 Miyamaedaira, Miyamae-ku, Kawasaki-shi, Kanagawa 216-0006, Japan; Jun_Mori@Instron.com

**Keywords:** biocompatible titanium alloy, artificial hip joints, hot forging, selective laser melting, microstructure, tensile property, fatigue property, durability, custom-made implant

## Abstract

We investigated the microstructures, tensile properties, fatigue strengths, and durability limits of hot-forged Ti-15Zr-4Nb (Ti-15-4) alloy artificial hip stems. These properties were compared with those of Ti-15Zr-4Nb-4Ta (Ti-15-4-4) and Ti-6Al-4V (Ti-6-4) alloy stems annealed after selective laser melting. The tensile and fatigue properties of test specimens cut from Ti-15-4 stems annealed after hot forging were excellent compared with those of the Alloclassic Zweymüller Stepless (SL) stem, which is used globally. The 0.2% proof stress (σ_0_._2%PS_), ultimate tensile strength (σ_UTS_), total elongation (TE) at breaking, and fatigue strength (σ_FS_) after 10^7^ cycles were 919 ± 10, 983 ± 9 MPa, 21 ± 1%, and 855 ± 14 MPa, respectively. The durability limit (P_D_) after 5 × 10^6^ cycles of Ti-15-4 stems was excellent compared with that of the SL stem. The σ_UTS_ values of 90°- and 0°-direction-built Ti-15-4-4 rods were 1032 ± 1 and 1022 ± 2 MPa, and their TE values were 14 ± 1% and 16 ± 1%, respectively. The σ_FS_ values of annealed 90°-direction-built Ti-15-4-4 and Ti-6-4 rods were 640 ± 11 and 680 ± 37 MPa, respectively, which were close to that of the wrought Ti-15-4 rod (785 ± 17 MPa). These findings indicate that the hot forging and selective laser melting (SLM) techniques can also be applied to the manufacture of artificial hip prostheses. In particular, it was clarified that selective laser melting using Ti-15-4-4 and Ti-6-4 powders is useful for the low-cost manufacturing of custom-made artificial joint prostheses and other prosthetic implants.

## 1. Introduction

Titanium (Ti) alloys widely used in artificial hip joints and various other prosthetic implants contain aluminum (Al) and vanadium (V); Ti and Al ions are both cytotoxic. The high cytotoxicity of V ions has been a concern because pseudocarcinoma was found to develop owing to exposure to metal-wear powder generated from the sliding part of a metal-on-metal hip joint. Moreover, ocular inflammation caused by Al ions released from fine particles adhering to the intraocular lens has been reported [1]. With these as a background, the development of prosthetic implants using Ti alloys that are safe and reliable even after biological implantation over a long term has been a clinical issue. To obtain basic data required for the development of orthopedic Ti-Zr alloy implant devices with excellent biocompatibility and osseointegration, biological safety evaluation tests of three Ti-Zr alloys (Ti-15Zr-4Nb, Ti-15Zr-4Nb-1Ta, and Ti-15Zr-4Nb-4Ta) in accordance with the ISO 10993 series were performed under both normal and accelerated extraction conditions. The biological safety evaluation tests of these three Ti-Zr alloys in accordance with the ISO 10993 series showed no negative effect of either normal or accelerated extraction. Moreover, we examined the maximum pullout properties of grit-blasted Ti-15Zr-4Nb (Ti-15-4) after its implantation in rabbits. The surface roughness (RA) and maximum pullout load of the Ti-15-4 alloy grit-blasted with 24-grit Fuji Random WA Al_2_O_3_ particles were the same as those of the grit-blasted Alloclassic Zweymüller Stepless (SL) stem surface. The area ratios of Al_2_O_3_ particles of these two materials were also similar [1]. Thus, the grit-blasted Ti-15-4 alloy could be used for artificial hip joint stems. Ti-15-4, designated as an α-β-type Ti alloy, has been developed as a highly biocompatible alloy for long-term biomedical application [2], as well as T 7401-4, in accordance with Japanese Industrial Standards (JIS) [3]. We focused on hot forging and molding to manufacture artificial hip stems with Ti-15-4 round bars using the approved Alloclassic SL stem for comparison.

There have been many studies in which the durability of artificial hip stems was evaluated by finite element analysis (FEM) simulation in accordance with the ISO 7206 series [4,5,6,7,8,9,10,11,12,13,14,15,16,17,18,19,20]. Also, additive manufacturing (AM) is expected to be a new technology for manufacturing biocompatible Ti orthopedic implants such as custom-made implants. There have been many studies on the mechanical and microstructural properties of selective-laser-melted Ti-6Al-4V (Ti-6-4) alloys [21,22,23,24,25,26,27,28,29,30]. On the other hand, there are only a few reports on the durability of selective-laser-melted Ti-6-4 artificial hip stems [9].

In this study, we investigated the microstructure, tensile properties, and fatigue strengths of specimens cut from artificial hip stems annealed at 700 °C for 2 h after hot forging. The mechanical properties of hot-forged Ti-15-4 hip stems were compared with those of the approved Alloclassic SL stems. To develop artificial hip prostheses with high durability, the durability limits of hot-forged Ti-15-4 hip stems were also compared with those of the approved product Alloclassic SL stem. Moreover, to compare the microstructure, tensile properties, fatigue strengths, and durability limits of hip stems obtained by hot forging and selective laser melting, we manufactured artificial hip stems by 3-D layer manufacturing with Ti-15Zr-4Nb-4Ta (Ti-15-4-4) and Ti-6-4 powders. In particular, we focused on the fatigue properties of selective-laser-melted and hot-forged Ti alloys. The results obtained in this study are expected to be useful for the development of biocompatible Ti alloy artificial hip prostheses and a low-cost manufacturing process.

## 2. Experimental Procedure

### 2.1. Test Specimens

Alloclassic Zweymüller stepless (SL) stems (Ti-6Al-7Nb (Ti-6-7), Zimmer Biomet, Tokyo, Japan), which are used globally and show excellent long-term clinical results, were selected as the model hip stem implants for mold hot forging and selective laser melting. Size 01 in the catalog (2839, stem length, 135 mm; size S in this study), size 4 (2844, stem length, 160 mm; size M), and size 7 (2847, stem length, 170 mm; size L) were used for comparison.

#### 2.1.1. Hot Die Forging of Artificial Hip Stems

We established the conditions for rolling Ti-15-4 alloy billets (100 mm square) into rods (e.g., 22 and 25 mm in diameter) with optimal shapes for the high-temperature forging of artificial hip joint stems [2]. With the β-transus temperature (T_β_, 850 °C) used as a reference, hot rolling was started at a temperature of T_β_–50 °C. Ti-15-4 billets were continuously hot-rolled and shaped into rods with a diameter of 22 or 25 mm. The round bars (wrought Ti-15-4 alloy) were annealed at 700 °C for 2 h. Details of the feedstock material and manufacturing process for the Ti-Zr alloy are described in Ref. [2].

Figure 1 shows the hot die forging for manufacturing cementless artificial hip stems. Ti-15-4 rods with a diameter of 22 or 25 mm were shaped into artificial hip stems by die forging at a high temperature. Table 1 shows hot forging conditions and sizes of hot-forged stems. We manufactured three types of mold, large (size L), medium (size M), and small (size S), for forging artificial stems with the same shape as the approved product Alloclassic SL stems. The mold is a set of two parts, the upper and lower parts. Considering the forging ratio at places where burrs are frequently generated, the swaging technique was used to shape a Ti-15-4 rod to be processed into a spindle to reduce the quantity of generated burrs. The Ti-15-4 rod was shaped into a spindle so that the forging ratio ((cross-sectional area after forging)/(cross-sectional area before forging)) was 1.5–2.0. The spindle-shaped Ti-15-4 specimens were continuously introduced into a high-frequency continuous-heat-treatment furnace, and die forging was started at a temperature of 740 or 780 °C (mainly 780 °C). The spindle-shaped Ti-15-4 specimens were subjected to bending, rough forging, deburring, and finish forging to obtain three types of artificial hip stem with different sizes: S, M, and L. The oxidized layer formed on the surface during hot forging was removed by blasting and pickling after annealing at 700 °C for 2 h. The hip stems were blasted to have an RA of approximately 3 to 4 µm using high-purity Al_2_O_3_ blasting medium [1]. This was similar to the RA of the approved product Alloclassic SL stems.

#### 2.1.2. Selective Laser Melting of Artificial Hip Stems and Rod Specimens

Ti-15-4-4 alloy powder was prepared by LPW Technology Ltd. (Cheshire, United Kingdom). Ti-15-4-4 and Ti-6-4 (EOS GmbH Electro Optical System, Krailling, Germany) powders were prepared by plasma atomization. Figure 2 shows the particle size distribution of the Ti-15-4-4 powder. Figure 2 also shows the D_10_, D_50_, and D_90_ particle sizes corresponding to 10%, 50%, and 90% of the cumulative distribution, respectively. The D_10_, D_50_, and D_90_ distribution of the Ti-15-4-4 powder showed the same tendency as that of the commercially pure Ti grade 2 powders and Ti-6Al-4V alloy powders [21,28,31].

The Ti-15-4 and Ti-6-4 powders were selective-laser-melted in Ar atmosphere using a system comprising an EOS M290 machine (EOS GmbH Electro Optical System, Krailling, Germany), EOSPRINT v. 1.5 (EOS GmbH Electro Optical System, Krailling, Germany) and HCS v. 2.4.14 software (EOS GmbH Electro Optical System, Krailling, Germany), and the Ti64 Performance M291 1.10 parameter set. The laser beam power (P) and the hatch spacing between scan passes (H) were 280–300 W and 0.13–0.14 mm, respectively. The laser scan speed (V) and powder stacking (deposited layer) thickness (T) were fixed from 1200 to 1300 mm/s and 0.03 mm, respectively. The laser spot focus diameter was 0.1 mm. The volumetric energy density (E) = P/(H·T·V) [23,26] was approximately 60 J/mm^3^.

Artificial hip stems and cylindrical rods (diameter, 9 mm; height, 50 mm) built by selective laser melting were cut from the support materials. The building direction of the cylindrical specimens was set to 0° (hereafter, 0° direction) and 90° (90° direction) for the base plate using the Ti-15-4-4 powders. The Ti-15-4 stems and cylindrical rods after selective laser melting were heat-treated at 760 °C for 4 h followed by air cooling. For comparison, Ti-6-4 stems and Ti-6-4 rods were similarly selective laser melted. The selective-laser-melted Ti-6-4 specimens were annealed at 840 °C for 4 h followed by air cooling.

#### 2.1.3. Chemical Analyses of Test Specimens

The chemical compositions of hot-forged Ti-15-4 and selective-laser-melted Ti-15-4-4 and Ti-6-4 femoral stems are shown in Table 2. The inert gas fusion thermally conductive detection method was used for the analysis of H and N, and the inert gas fusion infrared absorption method was used for the analysis of O. The combustion-infrared absorption method was used for the analysis of C, and inductively coupled plasma emission spectroscopy (ICPES) was used for the analysis of Zr, Nb, Ta, Pd, Al, V, and Fe. These measurements were carried out in accordance with JIS H 1619 [32], JIS H 1612 [33], JIS H 1620 [34], JIS H 1617 [35], and ASTM E 2371 [36]. The chemical analyses were performed at Kobelco Research Institute, Inc. (Hyogo, Japan). The difference in metal concentration between powder and selective-laser-melted alloy was very small for Ti-15-4-4 and Ti-6-4 alloys.

### 2.2. Microstructural Observation

The microstructure of each specimen after annealing was observed by optical microscopy, scanning electron microscopy (SEM), and transmission electron microscopy (TEM). Each test specimen was embedded in resin and polished to a mirrorlike finish with 200–4200 grit waterproof emery paper and an oxide polishing (OP-S) suspension. Then, each test specimen was etched with nitric acid solution containing 3 vol% hydrogen fluoride. The microstructures were analyzed by optical microscopy (ECLIPSE LV150, Nikon, Tokyo, Japan), SEM (Quanta 200 FEG, Philips, Tokyo, Japan; acceleration voltage, 15 kV), and field emission TEM (FE-TEM, JEM-2300T, JEOL, Tokyo, Japan; acceleration voltage, 200 kV) with energy dispersive X-ray spectroscopy (JED-2300T, JEOL, Tokyo, Japan). For TEM, disk-shaped specimens of 3 mm diameter were prepared by electrolytic polishing with 5 vol% perchloric acid +60 vol% methanol +35 vol% butanol solution at 30 V and 46 mA at −30 °C. After electrolytic polishing, the transverse cross-sectional structure was observed by TEM at magnifications of 15,000× and 60,000×.

### 2.3. Evaluation of Mechanical Properties

#### 2.3.1. Room Temperature Tensile Tests

Each of the five uniform rod specimens shown in Figure 3b (rod diameter, 3 mm; gauge length (GL), 15 mm) was cut from the hip stem at the position shown in Figure 3a. Also, five uniform rod specimens were cut from selective-laser-melted rod specimens (diameter, 9 mm; height, 50 mm). Room temperature tensile tests were carried out in accordance with JIS Z 2241 [37]. The tensile test specimens were pulled at a crosshead speed of 0.5% of the GL/min until the proof stress reached 0.2%. The crosshead speed was then changed to 3 mm/min and maintained at this value until the specimen fractured. The σ_0_._2%PS_, σ_UTS_, TE, and RA were measured in tensile tests. The mean and standard deviation were calculated from the results of five specimens.

#### 2.3.2. Fatigue Tests

Fatigue tests were conducted at room temperature in accordance with JIS T 0309 [38]. Miniature hourglass-shaped rod specimens (3 mm in minimum diameter and 50 mm in total length, as shown in Figure 3c) cut from hip stems at the position shown in Figure 3a and cylindrical rods were used for fatigue tests. The fatigue tests were carried out with a sine wave at a stress ratio R ((minimum cyclic stress (σ_min_)/(maximum cyclic stress (σ_max_)) of 0.1 and a frequency of 15 Hz in air. To obtain profiles of the relationship between σ_max_ and the number of cycles to failure N (S–N curves), the specimens were subjected to cycling at various constant maximum cyclic loads up to N = 10^7^ cycles, at which the specimens remained intact. The fatigue strength after 10^7^ cycles (fatigue limit, σ_FS_) was determined from the S–N curves. The fatigue limit and standard deviation were calculated from the results of at least 20 specimens.

#### 2.3.3. Durability Tests of Artificial Hip Stems

To obtain the profiles of maximum load vs. the number of cycles (P–N curves), durability tests of artificial hip stems were conducted in accordance with ISO 7206-4 third edition [39]. The artificial hip stem was fixed at the vertical distance from the center of the head to the upper level of the fixation, and at angles of α and β, as shown in Figure 4. The vertical distance (D) was set to 80 mm for the artificial hip stem with a length of 120 to 250 mm. In ISO 7206-4 second edition, D was 0.4 × CT, which was the distance (stem length, CT) between the center of the head (C) and the tip of the stem (T). The hot-forged Ti-15-4 stem was subjected to a durability test at D = 80 mm. At this D, it was assumed that the cement was loose in a cement-type stem [4,11]. The portions of clinical fixation of cementless stems are based on the Gruen Zone classification, namely, proximal, mid, and distal portions [40]. For a cementless Alloclassic SL stem (stem design with a tapered rectangular cross-sectional fixation), the stem is fixed by the fixing force of the autologous bone from the proximal portion to the distal portion [41]. Therefore, for the selective-laser-melted Ti-15-4-4 and Ti-6-4 stems, considering that the stems were of the cementless type and small, D = 0.4 × CT. α (in adduction), which is the angle between the load axis and the stem axis, was 10°, and β (in flexion), which is the angle between the line from the center of the head to the tip of the stem and the longitudinal-sectional stem axis when viewed from the back, was 9° [4,6,9,10,11,12]. The durability tests were carried out with a sine wave at a load ratio ((minimum cyclic load (P_min_)/(maximum cyclic load (P_max_)) of 0.1 and a frequency of 3 Hz in air. The durability limits after 5 × 10^6^ cycles (durability limit, P_D_) were determined from the P–N curves. The durability limit and standard deviation were calculated from the results of at least 12 specimens.

The durabilities of artificial hip femoral stems made of Ti-6-7 and Ti-6-4 alloys, which are globally used in clinical settings, were investigated for comparison with that of the present Ti-Zr alloy in this study. The cementless total hip stems used were Alloclassic Zweymüller SL, HA−TCP Fiber Metal Taper (65-7662-009-00; Ti-6-4; size 9; proximal diameter, 9 mm; stem length (CT), 120 mm; distal diameter, 6 mm; Zimmer Biomet, Tokyo, Japan), and S-ROM (A) (9005-23-210; Ti-6-4; CT, 115 mm; proximal diameter, 12 mm; distal diameter, 6 mm; Johnson & Johnson, Tokyo, Japan) stems.

### 2.4. Static Immersion Test

Dilute hydrochloric acid physiological saline (0.9%NaCl + HCl) solution adjusted to pH 2 with hydrochloric acid (HCl) is specified in ISO 16428 [43] as an accelerated extraction solution for the evaluation of corrosion resistance. Immersion tests were conducted in accordance with JIS T 0304 [44]. The 11 plate specimens, each with dimensions of 20 mm × 20 mm × 1 mm (thickness), were cut from the hot-forged Ti-15-4 and selective-laser-melted Ti-15-4-4 hip stems. The surface of each plate was polished with 1000-grit waterproof emery paper. The accelerated solution (0.9%NaCl + HCl, pH 2) was prepared as follows. Hydrochloric acid (1 mol/L) was added to physiological saline (0.9%NaCl) solution and the mixture was adjusted to pH 2 (0.9%NaCl + HCl solution). All test specimens were ultrasonically cleaned in ethanol. The 11 plate specimens were immersed for 7 d in the 0.9%NaCl + HCl solution (immersion rate, 3 cm^2^/1 mL) at 37 ± 1 °C. Blank extracts were similarly prepared but without the Ti-15-4-4 alloy plate. After immersion, the concentrations of Ti and alloying elements (Zr, Nb, and Ta) released into the solution over 7 d were determined (ng/mL) by inductively coupled plasma mass spectrometry (ICP-MS, NexION 300D, PerkinElmer, Kanagawa, Japan; isotopic mass numbers of Ti, Zr, Nb, and Ta, 49, 90, 93, and 181, respectively). An internal standard solution of Y (isotopic mass number, 89) was used for the correction of metallic concentrations. The amounts of Ti and alloying elements released (µg/cm^2^/week) were calculated.

### 2.5. Statistical Analysis

The mean and standard deviation of tensile properties were calculated from the results of five specimens. The S–N curve, the fatigue limits of hot-forged Ti-15-4 and selective laser-melted Ti-15-4-4 and Ti-6-4, and the standard deviation were calculated with statistical analysis software based on JSMS-SD-06-08 [45]. The fatigue limit and standard deviation were calculated in accordance with DIN 50100 [46] and ASTM E739 [47]. With the statistical analysis software used in this study, the fatigue/durability limit and standard deviation can be calculated when there are eight or more specimens and two or more specimens that did not break after 5 million cycles or more.

## 3. Results and Discussion

### 3.1. Microstructures and Mechanical Properties of Hot-Forged Artificial Hip Stems

#### 3.1.1. Microstructures of Hot-Forged Artificial Hip Stems

Figure 5 shows (a) optical microscopy and (b) SEM images of the transverse (T) section near the center at the 80 mm position from the stem head of the annealed Ti-15-4 stem (size S) after hot forging. As shown in Figure 5b, the β (beta)-phase (body-centered cubic structure, bcc) appears white in the SEM image. In the optical microscopy and SEM images of the annealed Ti-15-4 hip stem, the β-phase that precipitated in the grain boundaries of the α (alpha) (hexagonal-close-packed structure, hcp) matrix [2] was found to be produced by hot forging. Figure 5d,e shows TEM images of the T sections. Figure 5e shows a comparison between the calculated and measured interplanar distances (d) and Miller indices. There was good agreement between both values. In the calculation of d values for the β-phase (bcc) and α-phase (hcp), the lattice parameters a = b = c = 0.331 nm for bcc Ti (International Centre for Diffraction Data (ICDD) No. 044-1288) and a = b = 0.295 and c = 0.468 nm for α Ti (ICDD No. 044-1294) were used. From the results of electron beam diffraction analysis, the β-phase was found to be precipitated in the grain boundaries of the hcp matrix. The microstructure of hot-forged Ti-15-4 was finer than that of the Alloclassic Zweymüller SL stem (Ti-6-7), as shown in Figure 5c. Similar microstructures were obtained for M and L stems.

#### 3.1.2. Mechanical Properties of Hot-Forged Artificial Hip Stems

Figure 6 shows a comparison between the P–N curves of Ti-15-4 alloy hip stems (sizes S and M) and Alloclassic SL stems. A durability test was carried out for more than 5 million cycles in accordance with ISO 7206-4 criteria. The durability limits (P_D_) after 5 million cycles were 3400 ± 495 N for an S stem and 6800 ± 606 N for an M stem. The durability limits of the Ti-6-7 alloy stem were 3000 ± 512 N for the S stem and 6400 ± 463 N for the M stem. It was found that the Ti-15-4 hip stem hot-forged using the forging technology developed in this study had a durability limit higher than that of the Alloclassic SL stem. It fully satisfied the durability limit at 5 million cycles of 2300 N specified in ISO 7206-4 third edition. From these results, the artificial hip joint stem manufactured using the hot forging technology developed in this study is expected to be used clinically. Since the durability limit of the M stem was considerably high, we decided to mainly investigate the mechanical properties of the S stem thereafter. The standard deviation (SD) for the mean value of P_D_ in this study was calculated using data of the entire P−N curve, and it was assumed that the SD was distributed to the same extent even for P_D_. The ratio of SD to the mean value of PD (SD/mean P_D_, 495/3400 = 0.15, 606/6800 = 0.09, 512/3000 = 0.17, 463/6400 = 0.07) was in the range of 7−17%. This SD/mean P_D_ ratio for hip stems tended to be larger than the SD/mean σ_FS_ ratio shown in Table 3. This is considered to be due to the torsional force applied in addition to the compressive load in the durability test of the stems. The fatigue fracture from the edge may be related to these load and force factors.

Miniature mechanical specimens were cut from the hot-forged hip stems and subjected to tensile tests at room temperature and fatigue tests up to 10^7^ cycles. Table 3 shows the tensile properties (*n* = 5, mean ± standard deviation) of miniature specimens cut from the Ti-15-4 hip stems annealed at 700 °C for 2 h after hot forging at 780 or 740 °C. The tensile strength of the hot-forged stem tended to be higher than that of the 22 or 25 mm (wrought) Ti-15-4 rod before hot forging. Also, the tensile strength of the hot-forged stem at 780 or 740 °C was close to that of the Alloclassic SL (Ti-6-7) stem. It was close to the σ_UTS_ (977–985 MPa) obtained by the finite element analysis of the durability of the Alloclassic SL stem with Ti-6-4 alloy annealed at 700 °C after forging at 880–950 °C [7].

Figure 7 shows S–N curves of the hot-forged Ti-15-4 and Alloclassic SL stems, and the wrought Ti-15-4 rod. The fatigue strength of the Ti-15-4 stem hot-forged at 780 °C was ~855 MPa and slightly higher than that of the stem hot-forged at 740 °C, which was higher than those of the Alloclassic SL stem and wrought Ti-15-4 rod. The σ_FS_/σ_UTS_ (0.85) of the hot-forged Ti-15-4 was slightly higher than that of the Alloclassic SL stem (0.78). Thus, the fatigue strength of the hot-forged Ti-15-4 stem was higher than that of the Alloclassic SL stem. It is considered that this improvement in the fatigue strength of the hot-forged Ti-15-4 stem was attributable to its fine microstructure, as shown in Figure 5a.

### 3.2. Microstructures and Mechanical Properties of Selective-Laser-Melted Ti Alloy Hip Stems and Rod Specimens

#### 3.2.1. Microstructures of Selective-Laser-Melted Stems and Rod Specimens

Figure 8 and Figure 9 show optical microscopy, SEM, and TEM images of the T section of annealed Ti-15-4-4 and Ti-6-4 rods after selective laser melting (90° direction). The selective-laser-melted Ti-15-4-4 and Ti-6-4 rods had an acicular structure. TEM images of the selective-laser-melted Ti-15-4-4 and Ti-6-4 rods show that they consisted of a fine lath martensitic (α’) (hcp, lattice parameters a = b = 0.295, c = 0.468 nm) structure that precipitated with the fine β-phase (bcc, a = b = 0.331 nm) in the grain boundary of the α’ matrix that formed owing to rapid solidification. The values of these lattice parameters were consistent with those of the Ti material properties [48]. The liquidus temperature of the once-sintered Ti-15-4-4 rod measured by differential thermal analysis (DTA, TG–DTA 2200SA, Bruker Corp., Kanagawa, Japan) was 1653 °C. A similar acicular structure was found in many studies on selective-laser-melted Ti-6-4 alloys [21,22,23,24,25,26,27,28,29,30].

#### 3.2.2. Mechanical Properties of Selective-Laser-Melted Stems and Rod Specimens

The mechanical properties of the selective-laser-melted Ti-15-4-4 and Ti-6-4 rods are shown in Table 4. The tensile properties of the selective-laser-melted Ti-15-4-4 and Ti-6-4 rods were close to those of the hot-forged Ti-15-4 rod shown in Table 3. The mechanical strengths of the selective-laser-melted Ti rods were close to that of the wrought Ti-15-4 rod. The tensile properties of the selective-laser-melted Ti-15-4-4 and Ti-6-4 rods fully satisfied the tensile properties (σ_0_._2%PS_ ≥ 780, σ_UTS_ ≥ 860 MPa, and TE ≥ 10%) specified in JIS T 7401-4 and ISO 5832-3 [49]. Figure 10a shows the mechanical properties (σ_0_._2%PS_, σ_UTS_, TE, and RA) of the selective-laser-melted Ti-6-4 rods as a function of the number of repetitions of laser melting. Hardly any effect of the number of repetitions on σ_0_._2%PS_, σ_UTS_, TE, and RA was observed up to 10 repetitions. The reason was considered to be the negligible increase in the O concentration of the selective-laser-melted Ti-6-4 rods, as shown in Figure 10b.

The tensile properties shown in Table 4 showed similar values to those reported in the literature [21,22,26,27,28,29,30].

Figure 11 shows SEM images of the fracture surfaces of the hot-forged hip stem and selective-laser-melted Ti-15-4-4 and Ti-6-4 rods after the tensile test. Magnified images of the rectangular areas in Figure 11a,c,e are shown in Figure 11b,d,f, respectively. Dimples were observed on the fracture surfaces, as shown in Figure 11b,d,f. These surfaces with a dimple showed similar results to those reported in the literature [21,22,26,28].

Figure 12 shows the S–N curves of Ti-15-4-4 and Ti-6-4 rods annealed after selective laser melting. The fatigue strengths (σ_FS_) of the 90°-direction-built Ti-15-4-4 (once-melted) and Ti-6-4 (once- and 10-times-melted) rods were ~640, ~680, and ~660 MPa, respectively. The σ_FS_ and σ_eq_/σ_FS_ values of the selective-laser-melted Ti-15-4-4 and Ti-6-4 rods are shown in Table 4. In particular, it was found that the fatigue strengths of the selective-laser-melted Ti-15-4-4 and Ti-6-4 rods were close to that of the wrought Ti-15-4 alloy rod. The fatigue strengths (100–600 MPa) of the selective-laser-melted Ti-6-4 rods obtained in this study were higher than those reported in the literature [21,22,26]. The improvements in fatigue strength are effective in improving the lath martensitic (α’) structure and reducing the thermal stress caused by selective laser melting. To increase the fatigue strength of selective-laser-melted Ti alloys, it is necessary to improve the morphology of the α’ martensitic structure and change the α’ martensitic structure to a two-phase structure of α (hcp)–β (bcc) by heat treatment [25].

Figure 13 shows SEM images of the fracture surfaces of the hot-forged Ti-15-4 stem and the 90°-direction-built Ti-15-4-4 and Ti-6-4 rods. A fatigue crack developed with a fatigue fracture from the internal parts of the specimen, and striations were observed, as shown in Figure 13b,d,f. These fracture surfaces showed similar results to those reported in the literature [21,22,30].

Figure 14 shows the L–N curves of the selective-laser-melted Ti-15-4-4 and Ti-6-4 size S stems and the approved product HA−TCP and S−ROM stems. The durability limit of the selective-laser-melted Ti-15-4-4 stem was lower because the selective laser melting conditions developed for the Ti-6-4 alloy were used, and the selective laser melting conditions for the Ti-15-4-4 alloy are as yet not developed. On the other hand, the durability limit of the selective-laser-melted Ti-6-4 stem was ~2500, which was much higher than those of the approved product HA−TCP and S−ROM stems.

### 3.3. Stress Analysis of Artificial Hip Stem

The stress analysis of the durability test results of stems was performed using the fatigue strengths shown in Table 3 and Table 4. Tensile and compressive stresses increased toward the exterior (A) and interior (B) surfaces respectively from the center of the stem, as shown in Figure 4 [9,12,17]. Figure 15 shows the stress analysis of the durability test results of a fixed region of the A−B plane specified in ISO 7206-4 third edition [8]. Since the load F was inclined by 9° from the vertical direction, it could be decomposed into two components, F_1_ = F × cos9° and F_2_ = F × sin9°. As the neck angle of the stem was α (131°), the component force F_1_ could be decomposed into F_3_ = F_1_ × cos (180° − α + 10°) = F_1_ × cos (190° − α) and F_4_ = F_1_ × sin (190°). Moreover, F_1_ could be decomposed into two components (F_5_ = F_1_ × cos10° and F_6_ = F_1_ × sin10°); Mx = F_5_ × d_1_, Mx = F_6_ × d_2_, and My = F_2_ × d_2_.

When the compressive stress is positive, the net axial stress of the cross section, σ_z_, generated at position (*x*, *y*) on the A−B plane is given by
(1)$$\sigma_{Z}=\frac{F_{5}}{Cross-sectional\;area}+\frac{M_{x}}{I_{x}}y-\frac{M_{y}}{I_{y}}x,$$
(2)$$\sigma_{z}=\frac{F_{5}}{\mathrm{bh}}+\frac{F_{5}d_{1}-F_{6}d_{2}}{I_{x}}y-\frac{F_{2}d_{2}}{I_{y}}x.$$

Here, the inertia moments are *Ix* = b × h^3^/12 and *Iy* = h × b^3^/12. The shear stresses (*τ_zx_* and *τ_zy_*) generated in the x and y directions by the bending moment can be calculated using the following equations [4,8]:(3)$$\tau_{zx}=-\frac{Q_{y}F_{2}}{bI_{y}}\;\tau_{zy}=\frac{Q_{x}F_{6}}{hI_{x}},$$
(4)$$Q_{x}=\frac{b}{2}\left(\frac{h^{2}}{4}-y^{2}\right)\;Q_{y}=\frac{h}{2}\left(\frac{b^{2}}{4}-x^{2}\right).$$

Here, as shown in Figure 16, fatigue cracks are generated from the corners of the stem surface (x = b/2, y = −h/2 mm). Since $Q_{x}$ and $Q_{y}$ become zero at the material surface, the shear stresses (*τ_zx_* and *τ_zy_*) are zero.

Moreover, torque (*T*) is given by *T* = F_2_ × d_1_. The shear force generated by the torque is given by
(5)$$\tau_{zx}=-\frac{T}{2I_{x}}y\;\tau_{zy}=\frac{T}{2I_{y}}x.$$

The absolute values of the shear force generated by the bending moment and torque are used to calculate *τ_zx_* and *τ_zy_*, which are substituted into the following equation to determine the equivalent stress (σ*_eq_*) using the Von Mises criterion [8]:(6)$$\sigma_{eq}=[\sigma_{z}{}^{2}+3(\tau_{zx}{}^{2}+\tau_{zy}{}^{2}{)]}^{\frac{1}{2}}.$$

The equivalent stress (σ*_eq_*) can be used to directly compare the fatigue strengths shown in Table 3 and Table 4. Table 5 shows the maximum equivalent stress (σ*_eq_*) calculated for the hot-forged and selective-laser-melted Ti alloys with Equation (6). σ*_eq_* was calculated using the durability limits (x = 3.6 mm, y = −5.5 mm; 3400 N for Ti-15-4 and 3000 N for Alloclassic SL) of the S stems. The σ*_eq_* values of the Ti-15-4 and Alloclassic SL S stems were 871 and 791 MPa, respectively. These values were close to those (855 and 805 MPa) shown in Table 3. The σ*_eq_*/σ_FS_ values of the Ti-15-4 and Alloclassic SL S stems were 1.02 and 0.98, respectively, and a good match was obtained. This analysis is useful for developing artificial hip joints, identifying the worst specimens, and analyzing the durability test results of hip stems.

The σ*_eq_* values of the laser-melted Ti-15-4 and Ti-6-4 S stems were 107 and 178 MPa, respectively, as shown in Table 5. These values were considerably lower than those (640 and 680 MPa, respectively) shown in Table 3. The σ*_eq_*/σ_FS_ values of the laser-melted Ti-15-4 and Ti-6-4 stems were 0.17 and 0.26, respectively. With a load of 2300 N and the laser-melted Ti-6-4 stem fixed at D = 80 mm, the stems broke after around 100,000 cycles. To clarify the durability limit of 2300 N at fixation of 80 mm, it is necessary to consider the selective laser melting conditions of the Ti-6-4 hip stem. In addition, the stem is expected to be improved by examining the annealing conditions after the selective laser melting [25]. On the other hand, in the cementless type, the stem is fixed by the fixing force of the autologous bone from the proximal part to the distal part [42,50]. Even in the durability test with fixation D = 0.4 × CT (52 mm), the fixation position was clinically lower. Therefore, it is considered that selective laser melting can also be applied to custom-made artificial hip stems [4,9,51,52,53,54,55,56,57]. We performed a classical stress analysis in this study. It is expected that the results obtained in this study will be compared with those of finite element analysis [4,7,9,10,12,13,19,20].

### 3.4. Immersion Properties of Once-Laser-Melted Ti-15-4-4 and Wrought Ti-15-4 Plates

The amounts of Ti ions released from the once-melted Ti-15-4-4 and wrought Ti-15-4 plates were 1.70 and 1.73 μg/cm^2^/week, respectively. No significant difference in the amount of released Ti ions was seen between the laser-melted and wrought Ti-15-4 plates. Also, the release of alloying elements was not detected. Thus, the hot forging and AM of Ti alloys may be promising new manufacturing technologies.

## 4. Conclusions

We investigated the microstructures, tensile properties, and fatigue strengths of specimens cut from annealed artificial hip stems after hot forging. The mechanical properties of hot-forged Ti-15-4 hip stems were compared with those of the approved Alloclassic SL stem. The durability limits of the hot-forged Ti-15-4 hip stems were also compared with that of the approved product Alloclassic SL stem. Moreover, to compare the microstructures, tensile properties, fatigue strengths, and durability limits of hip stems obtained by hot forging and selective laser melting, we manufactured artificial hip stems by 3-D layer manufacturing using Ti-15-4-4 and Ti-6-4 powders.

For the hot-forged Ti-15-4 stem, it was found that the β-phase precipitated in the grain boundaries of the α (hcp) matrix was produced by hot forging. The microstructures were finer than that of the Alloclassic SL stem (Ti-6-7).The tensile strength of the Ti-15-4 stem hot-forged at 780 °C was close to that of the Alloclassic SL stem. The fatigue strength of the Ti-15-4 stem hot-forged at 780 °C was ~855 MPa, which was higher than that of the Alloclassic SL stem. The σ_FS_/σ_UTS_ (0.85) value of the hot-forged Ti-15-4 stem was slightly higher than that of the Alloclassic SL stem (0.78).The durabilities after 5 million cycles were 3400 ± 495 N for an S stem and 6800 ± 606 N for an M stem. The durabilities of the Alloclassic SL stem were 3000 ± 512 N for an S stem and 6400 ± 463 N for an M stem. The stress analysis of the durability test results of stems was performed using the fatigue strength (σ_FS_). The equivalent stress (σ*_eq_*) obtained by stress analysis was close to σ_FS_ obtained by fatigue tests of specimens cut from S stems.The selective-laser-melted Ti-15-4-4 and Ti-6-4 rods had an acicular structure. TEM images of the laser-melted Ti-15-44 and Ti-6l-4 rods showed a fine lath martensitic (α’) (hcp, a = b = 0.295, c = 0.468 nm) structure that precipitated with the fine β-phase (bcc, a = b = 0.331 nm) in the grain boundary of the α’ matrix. The durability limit of the laser-melted Ti-6-4 stem was ~2500, which was much higher than those of the approved product HA−TCP and S-ROM stems. It is considered that selective laser melting can also be applied to custom-made artificial hip stems.

## Figures and Tables

**Figure 1 materials-14-00732-f001:**
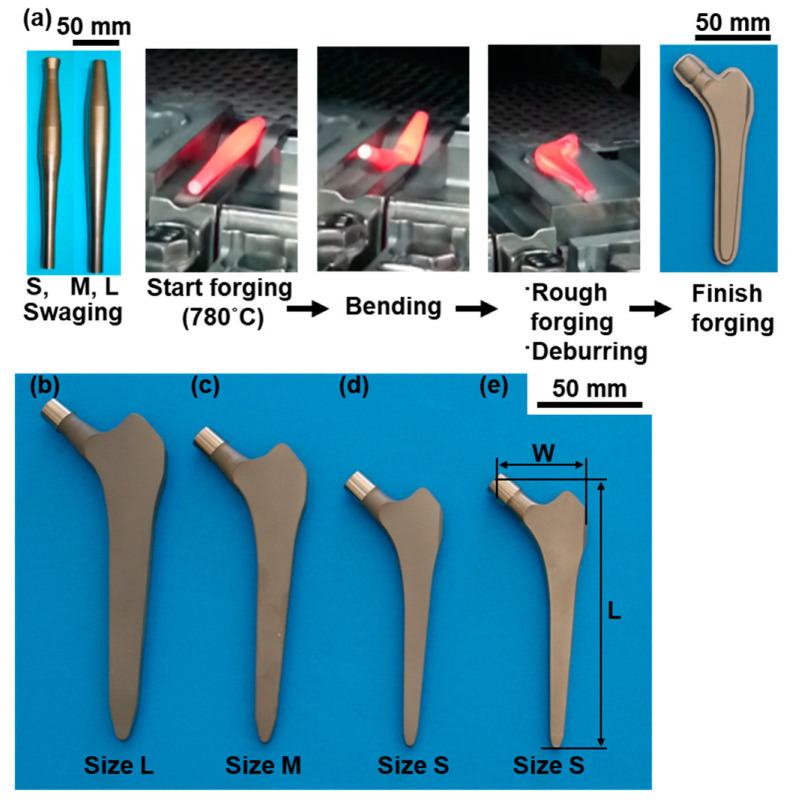
(**a**) Mold forging of artificial hip stem at high temperature, (**b**–**d**) hot-forged hip stems (sizes L, M, and S, respectively), and (**e**) laser-melted hip stem (size S).

**Figure 2 materials-14-00732-f002:**
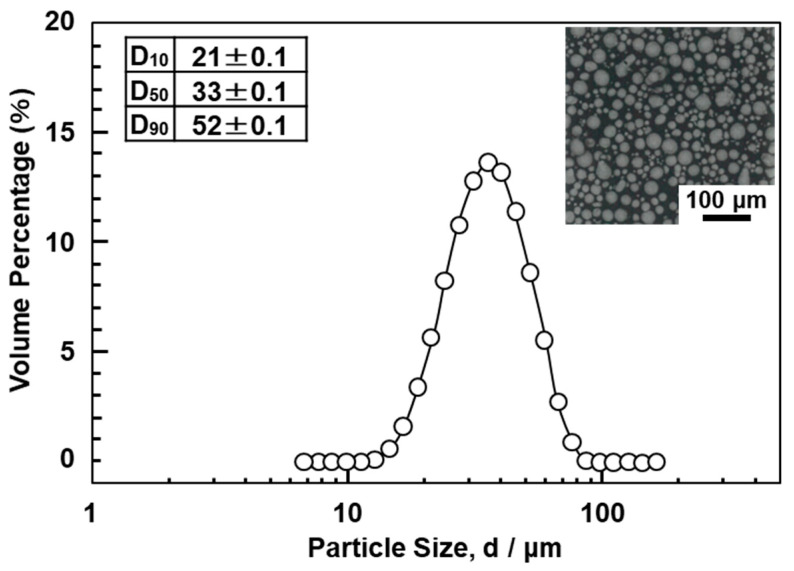
Particle size distribution of Ti-15Zr-4Nb-4Ta (Ti-15-4-4) powder.

**Figure 3 materials-14-00732-f003:**
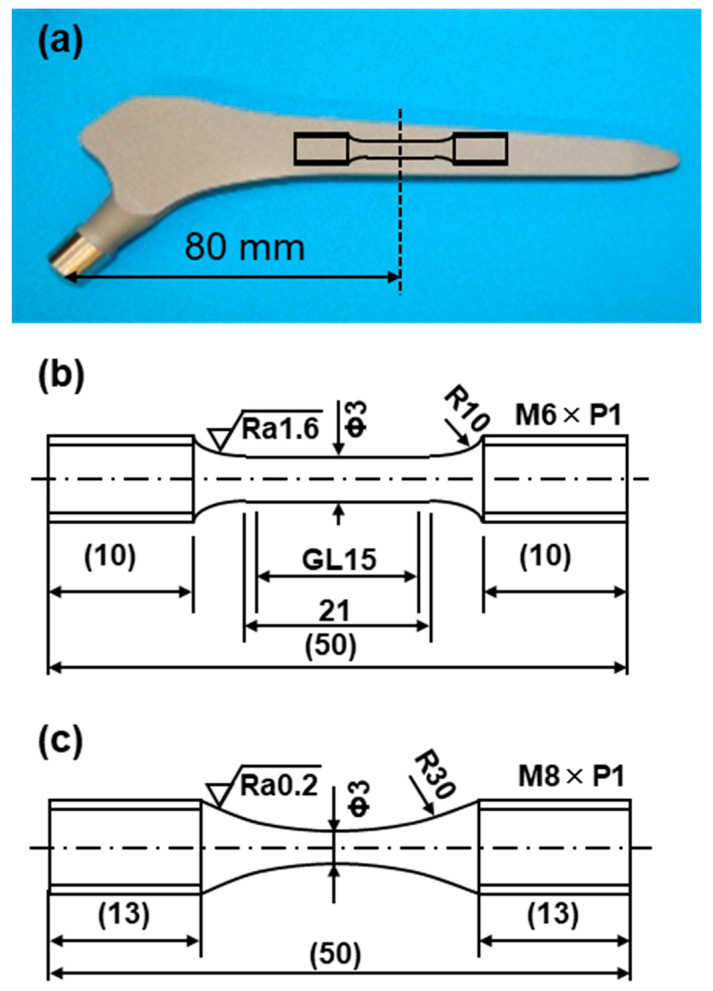
(**a**) Position of miniature specimen cut from hot-forged artificial hip stem; dimensions of specimens used for (**b**) room temperature tensile and (**c**) fatigue tests.

**Figure 4 materials-14-00732-f004:**
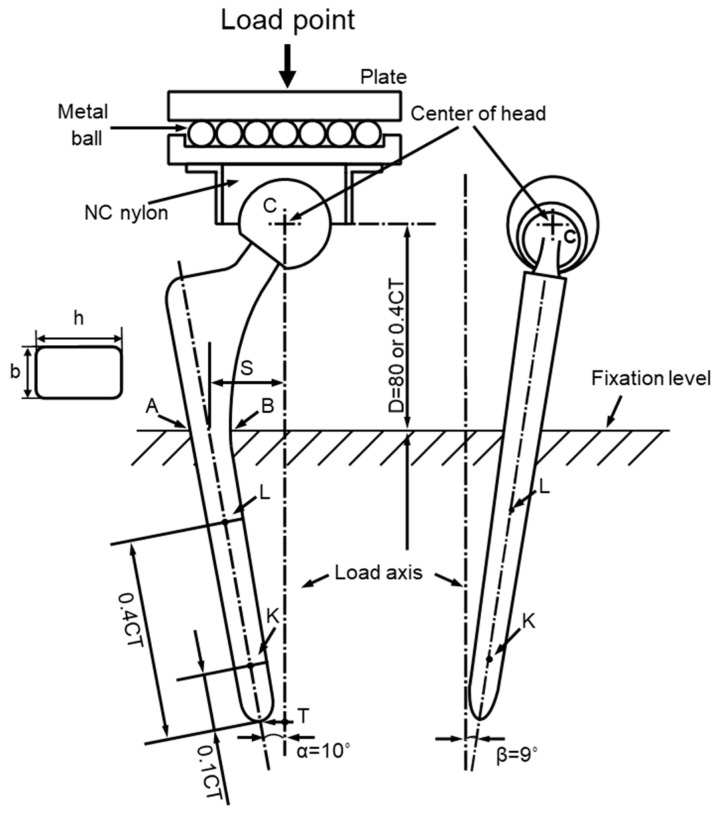
Fixation method in durability test using artificial hip stem in accordance with ISO 7206-4 [42].

**Figure 5 materials-14-00732-f005:**
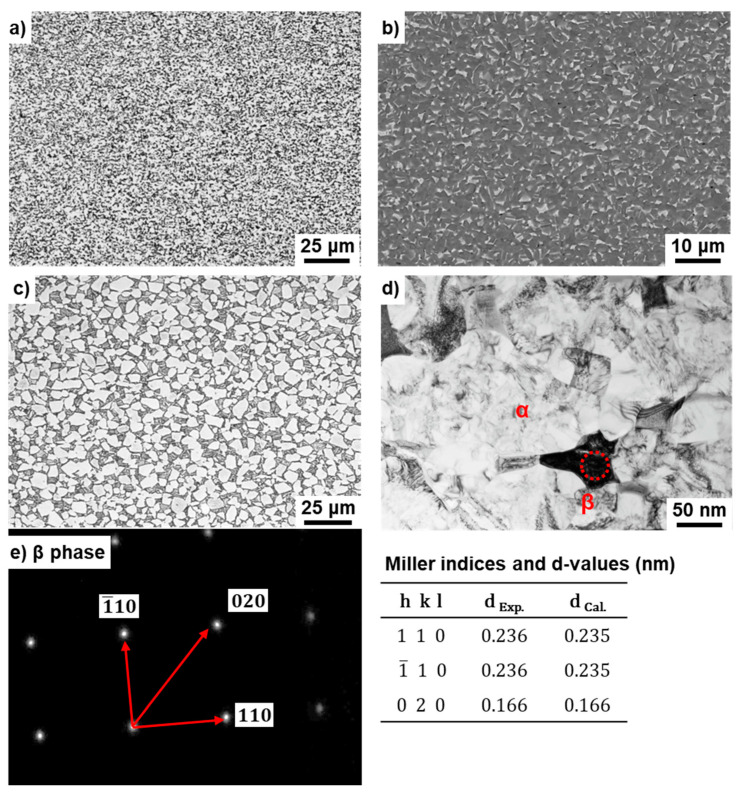
(**a**) Optical microscopy, (**b**) SEM, and (**d**) TEM images of Ti-15-4 stem hot-forged starting at 780 °C; (**e**) electron beam diffraction pattern at the encircled area in (**c**); (**c**) optical microscopy image of Alloclassic SL stem.

**Figure 6 materials-14-00732-f006:**
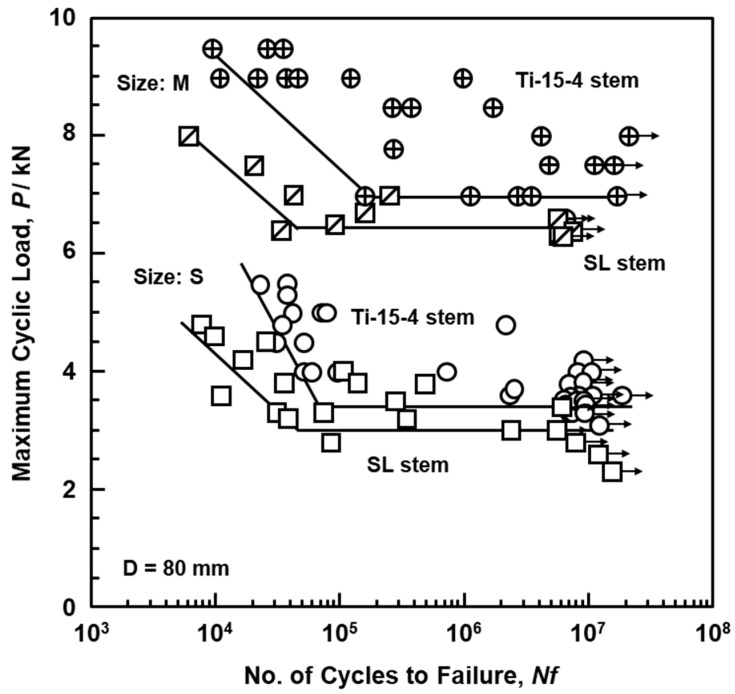
P–N curves of hot-forged Ti-15-4 hip stem (sizes S and M) and Alloclassic SL stem (sizes S and M) obtained from results of compression bending durability tests.

**Figure 7 materials-14-00732-f007:**
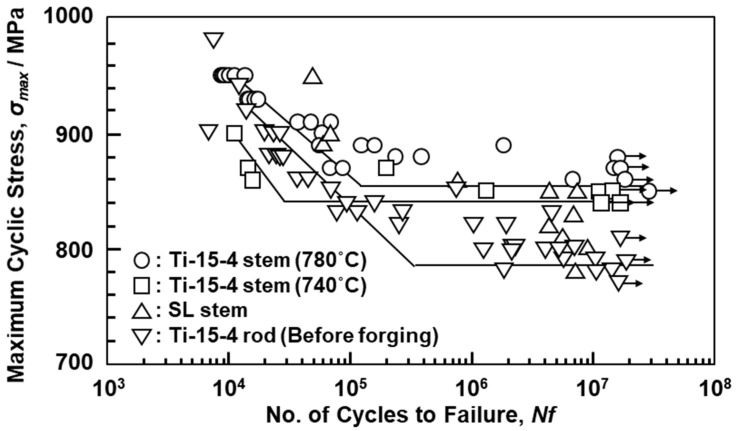
S–N curves of miniature cut from artificial hip stems.

**Figure 8 materials-14-00732-f008:**
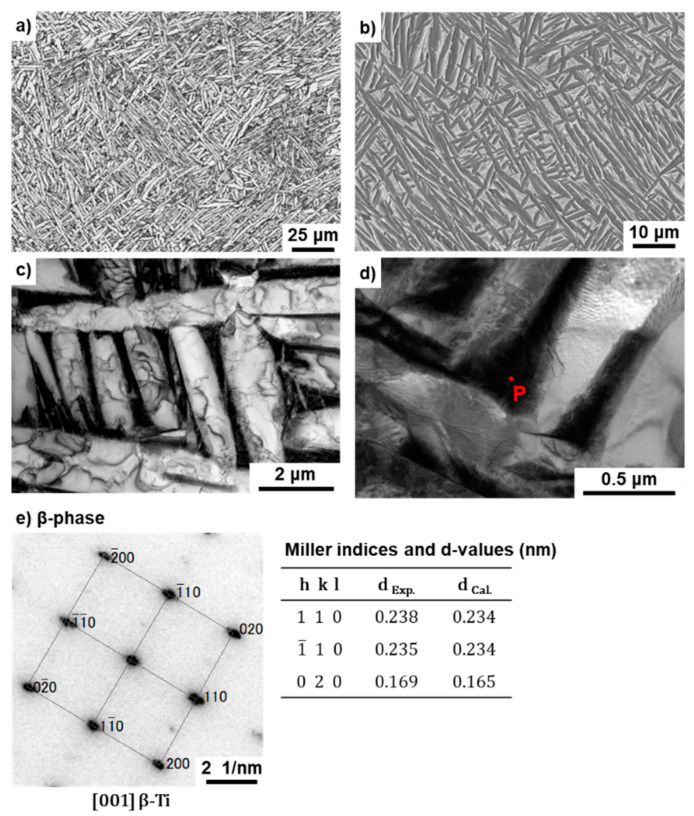
(**a**) Optical microscopy, (**b**) SEM, and (**c**,**d**) TEM images of selective-laser-melted Ti-15-4; (**e**) electron beam diffraction pattern obtained at the location indicated by P in (**d**) (precipitation).

**Figure 9 materials-14-00732-f009:**
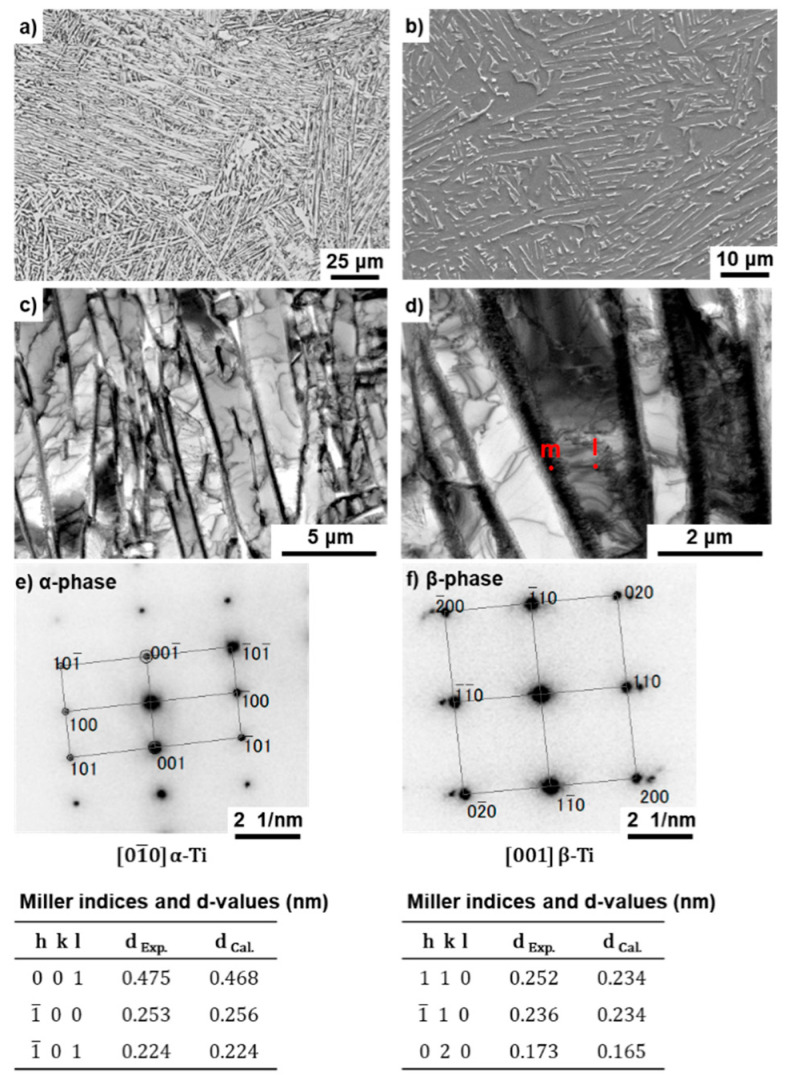
(**a**) Optical microscopy, (**b**) SEM, and (**c**,**d**) TEM images of selective-laser-melted Ti-6-4 rod; (**e**,**f**) electron beam diffraction patterns obtained at the location indicated by l and m in (**d**), respectively.

**Figure 10 materials-14-00732-f010:**
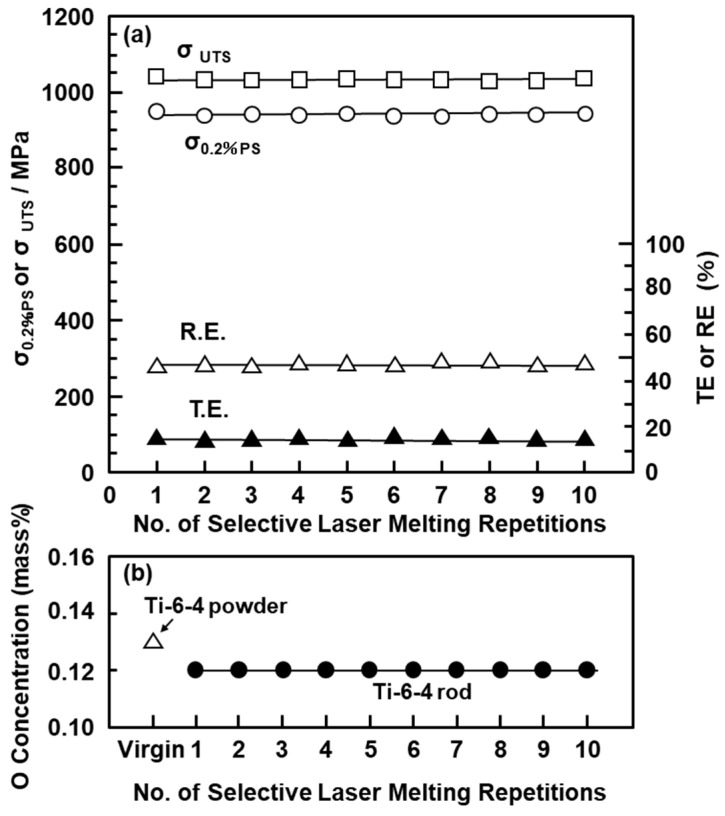
(**a**) Effects of repeated selective laser melting on (**a**) mechanical properties (σ_0_._2__%PS_, σ _UTS_, TE, and RA) of selective-laser-melted EOS Ti-6-4 rods built in 90° direction. (**b**) Effect of repeated selective laser melting on oxygen (O) concentration in Ti-6-4 powders and selective-laser-melted Ti-6-4 alloys.

**Figure 11 materials-14-00732-f011:**
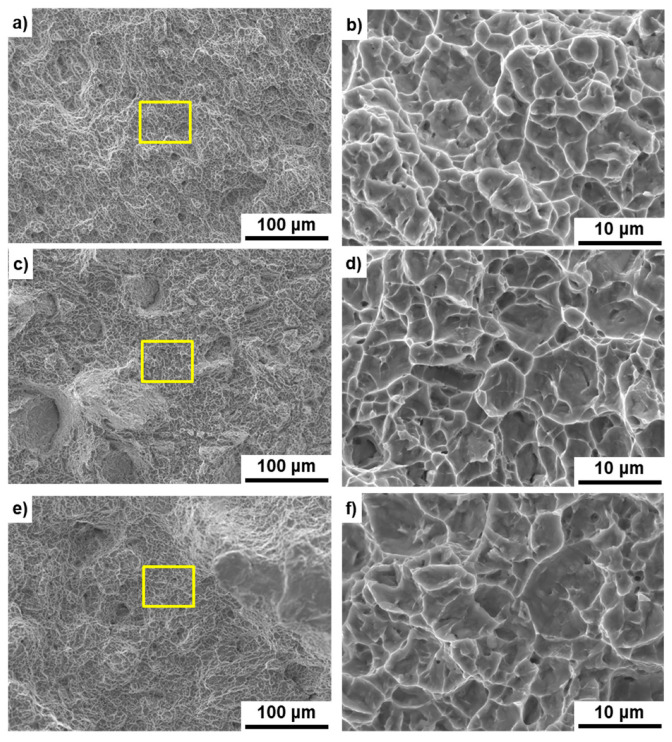
SEM images of fracture surface after tensile testing; (**a**,**b**) hot-forged hip stem, (**c**,**d**) selective-laser-melted Ti-15-4-4, (**e**,**f**) selective-laser-melted Ti-6-4; (**b**) magnification of rectangular area in (**a**), (**d**) magnification of rectangular area in (**c**), (**f**) magnification of rectangular area in (**e**).

**Figure 12 materials-14-00732-f012:**
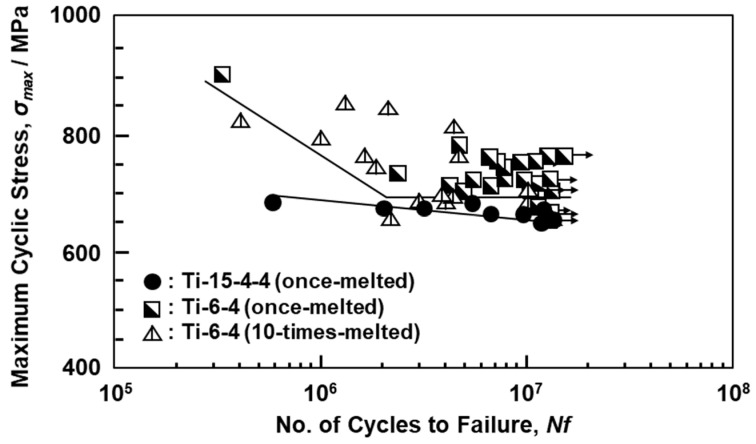
S–N curves of once-melted Ti-15-4 and once- and 10-times-melted Ti-6-4 rods built in 90° direction.

**Figure 13 materials-14-00732-f013:**
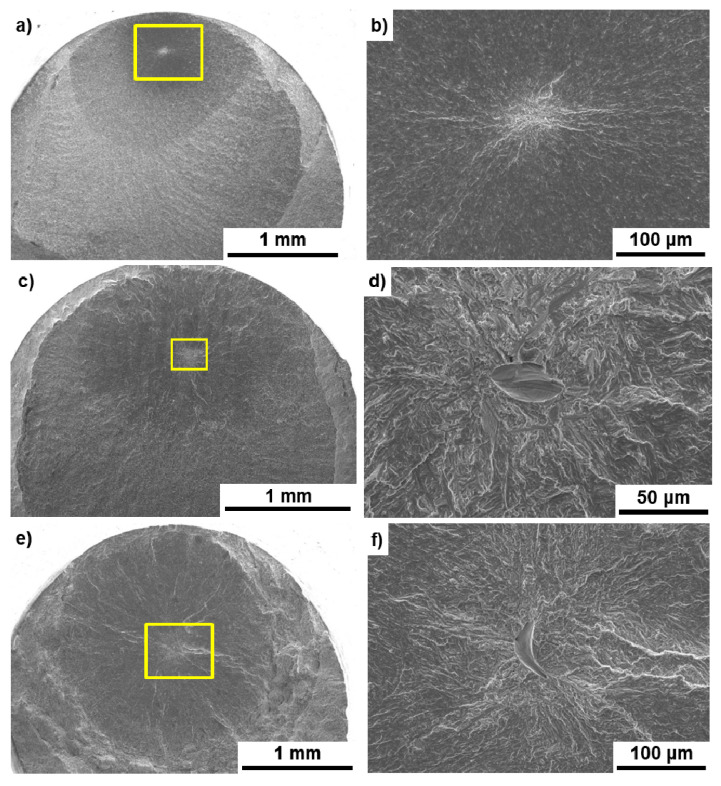
SEM images of fracture surfaces after fatigue testing; (**a**,**b**) hot-forged hip stem, (**c**,**d**) selective-laser-melted Ti-15-4-4, (**e**,**f**) selective-laser-melted Ti-6-4; (**b**) magnification of rectangular area in (**a**), (**d**) magnification of rectangular area in (**c**), (**f**) magnification of rectangular area in (**e**).

**Figure 14 materials-14-00732-f014:**
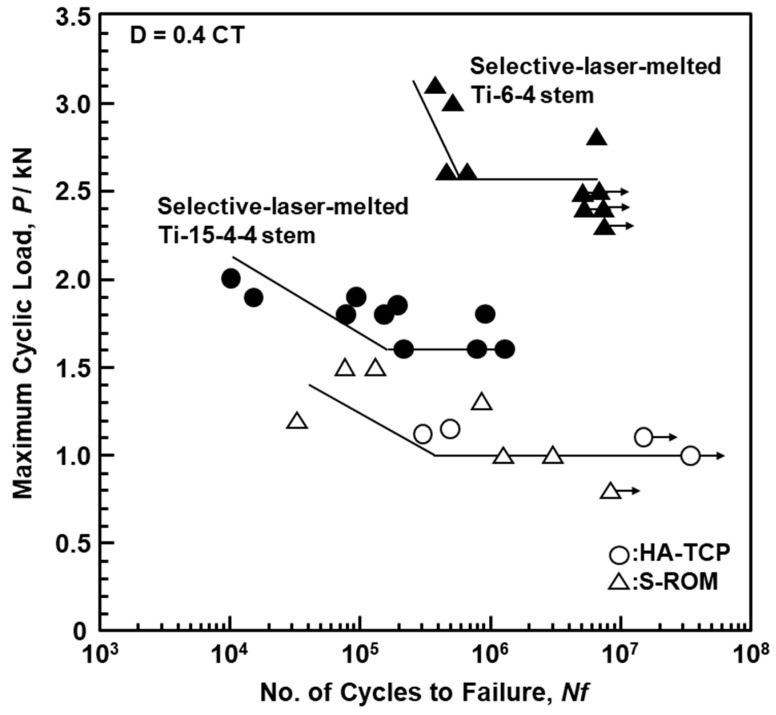
L–N curves of selective-laser-melted Ti-15-4-4 and Ti-6-4 hip stems (size S), approved product HA-TCP fiber metal, and S-ROM hip stems obtained by compression bending durability tests.

**Figure 15 materials-14-00732-f015:**
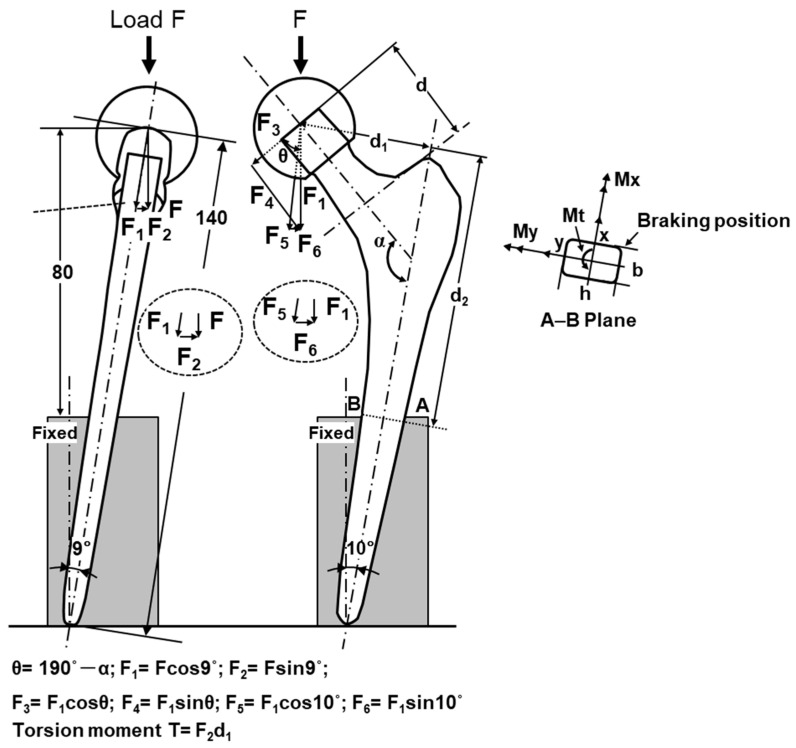
Stress analysis of durability test results in accordance with ISO 7206-4.

**Figure 16 materials-14-00732-f016:**
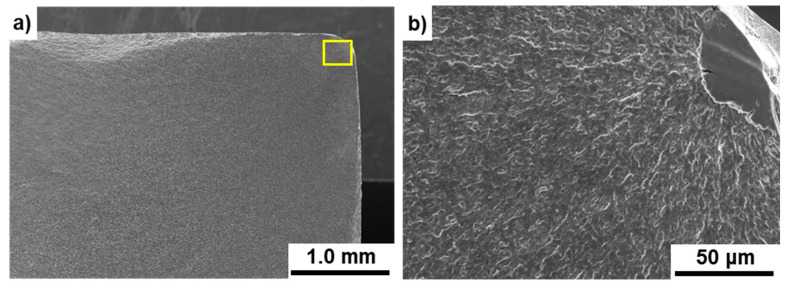
SEM images of fracture surfaces of durability-tested Ti-15-4 artificial hip stem; (**b**) magnification of rectangular area in (**a**).

**Table 1 materials-14-00732-t001:** Hot forging conditions and sizes of hot-forged stems.

Stem Size	Swaged Rod		Rough/Finish Forging		Forged Stem			
	Length mm	Max. Dia. mm	Starting Temp. °C		Length (L) mm	Width (W) mm	Number of Specimens	Ra μm
S	215	22	740	780	135	45	100	3–4
M	220	25	740	780	160	50	100	3–4
L	220	25	780		170	55	40	3–4

**Table 2 materials-14-00732-t002:** Chemical compositions (mass%) of hot-forged Ti-15Zr-4Nb (Ti-15-4) and selective-laser-melted Ti-15-4-4 and Ti-6Al-4V (Ti-6-4) femoral stems.

**Alloy**	**Zr**	**Nb**	**Ta**	**Pd**	**Fe**	**O**	**N**	**H**	**C**	**Ti**
Hot-forged Ti-15-4 stem	16.10	3.90	0.17	<0.01	0.026	0.254	0.080	0.001	0.010	Bal.
Ti-15-4-4 powder	16.6	3.97	3.12	<0.01	0.04	0.322	0.09	0.002	0.01	Bal.
Laser-melted Ti-15-4-4 stem	16.9	3.92	3.11	<0.01	0.04	0.34	0.096	0.003	0.009	Bal.
**Alloy**	**Al**	**V**	**Fe**	**O**	**N**	**H**	**C**	**Ti**		
Ti-6-4 powder	6.05	3.89	0.21	0.11	0.003	0.002	0.006	Bal.		
Laser-melted Ti-6-4 stem	6.25	3.98	0.19	0.089	0.02	0.0015	0.012	Bal.		

**Table 3 materials-14-00732-t003:** Tensile properties (0.2% proof stress (σ_0_._2%PS)_, ultimate tensile strength (σ_UTS)_, total elongation (TE), surface roughness (RA)), fatigue strengths after 10^7^ cycles (σ_FS_), and fatigue ratios (σ_FS_/σ_UTS_) of hot-forged Ti-15-4 stems.

Ti Alloy	σ_0_._2%PS_ /MPa	σ_UTS_ /MPa	TE (%)	RA (%)	σ_FS_ /MPa	σ_FS_ /σ_UTS_
Hot-forged stems						
Ti-15-4 rod (before forging)	887 ± 5	942 ± 2	20 ± 1	60 ± 1	785 ± 17	0.83
780 °C forged Ti-15-4 stem	919 ± 10	983 ± 9	21 ± 1	58 ± 2	855 ± 14	0.86
740 °C forged Ti-15-4 stem	912 ± 6	979 ± 7	19 ± 2	55 ± 5	840 ± 5	0.85
SL stem (Ti-6-7)	949 ± 23	1034 ± 23	16 ± 1	54 ± 1	805 ± 26	0.78

**Table 4 materials-14-00732-t004:** Tensile properties (σ_0_._2%PS_, σ_UTS_, TE, RA), fatigue strengths after 10^7^ cycles (σ_FS_), and fatigue ratios (σ_FS_/σ_UTS_) of selective-laser-melted Ti-15-4-4 and Ti-6-4 rods.

Ti Alloy	σ_0_._2%PS_ /MPa	σ_UTS_ /MPa	TE (%)	RA (%)	σ_FS_ /MPa	σ_FS_/σ_UTS_
Selective-laser-melted rods						
Once-melted 0° Ti-15-4	880 ± 2	1032 ± 1	14 ± 1	31 ± 2		
Once-melted 90° Ti-15-4	860 ± 3	1022 ± 2	16 ± 1	36 ± 7	640 ± 11	0.63
Once-melted 90° Ti-6-4	949 ± 3	1041 ± 2	15 ± 1	46 ± 2	680 ± 37	0.65
10-times-melted 90° Ti-6-4	946 ± 2	1036 ± 2	15 ± 1	47 ± 1	660 ± 14	0.64

**Table 5 materials-14-00732-t005:** Maximum equivalent stress, σ_eq_; coordinates (x, y ) of the location of σ*_eq_*; σ_FS_; and the ratio of maximum equivalent stress to fatigue limit of AB cross section.

Specimen	σ_eq_ /MPa	x, y /mm	σ_FS_ /MPa	σ*_eq_*/σ_FS_
Hot-forged Ti-15-4	871	(3.6, −5.5)	855	1.02
Alloclassic SL	791	(3.6, −5.5)	805	0.98
Laser-melted Ti-15-4-4	107	(3.6, −5.5)	640	0.17
Laser-melted Ti-6-4	178	(3.6, −5.5)	680	0.26

## Data Availability

Data sharing is not applicable.

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
