# Peer review of "Mechanical Performance of Artificial Hip Stems Manufactured by Hot Forging and Selective Laser Melting Using Biocompatible Ti-15Zr-4Nb Alloy"

_materials, 2021, doi:10.3390/ma14040732_

Round 1

Reviewer 1 Report

The authors have studied the mechanical performance of artificial hip stems manufactured by hot forging and laser sintering of Ti-15Zr-4Nb alloy. Some issues need to addressed as follows:

  1. Figure 1. The scale bar should be useful to get an idea of the different sizes of specimens. Figure (b-d) looks similar in size. Please check.
  2. A table would be better to present the different sized specimens and conditions.
  3. Figure 5- the diffraction image is poor. Please replace it.
  4. Section 3.1 and 3.3: The author refers the figure 1 in these sections. It makes the comparison difficult. It is better to mention the size or place the figure here.
  5. Micron marker absent in Figure 10e.
  6. Similarly, for figure 1e-f. The diffraction images are poor. Please improve indexing. The weaker intensity spots are hardly visible in figure 11d.
  7. Figure 7b is referred to near figure 14. It makes reading difficult. Please re-organize the figures for better readability.

Author Response

Thank you for the peer review of the manuscript, which has been revised in accordance with your comments.

The corrected parts are shown in red.

  1. Figure 1. We have added a scale bar. We took a picture of the stem again so that you can see its size.
  2. A table of stem sizes and hot forging conditions has been added as new Table 1.
  3. Figure 5. We have increased the resolution of the electron diffraction image and added calculations. A comparison between the measured and calculated d values was added.
  4. We have modified Section 3 as in the following sections to make it easier to read.

        3.1. Microstructures and Mechanical Properties of Hot-Forged Artificial Hip Stems.

              3.1.1. Microstructures of Hot-Forged Artificial Hip Stems

              3.1.2. Mechanical Properties of Hot-Forged Artificial Hip Stems

       3.2. Microstructures and Mechanical Properties of Selective-Laser-Melted Ti alloy Hip Stems and Rod Specimens

              3.2.1. Microstructures of Selective-Laser-Melted Stems and Rod Specimens

              3.2.2. Mechanical Properties of Selective-Laser-Melted Stems and Rod Specimens

        3.3. Stress Analysis of Artificial Hip Stem

        3.4. Immersion Properties of Once-laser-melted Ti-15-4-4 and Wrought Ti-15-4 Plates

       The text composition, tables, and figures have been rearranged in the order of hot forging, selective laser melting, and stress analysis.

  1. Figure 10(e). We have replaced the image of electron diffraction. We have also added a micron marker. A comparison between the measured and calculated d values was added.
  2. Figure 11. We have replaced the image of electron diffraction. We have also added a micron marker. A comparison between the measured and calculated d values was added.
  3. Figure 7b. We moved the new Figure 12 in the 3.2.2. (selective laser melting section) for better readability. The old Table 2 (b) has also been moved to new Table 4.

       We have corrected the section name of section 3, positions of figures and tables in the manuscript to make the  content easier to read.

Reviewer 2 Report

This is an experimental paper, dealing with a practical application of hot forging and metal additive manufacturing in biomedical engineering (namely the manufacturing and mechanical characterisation of a hip implant). This is a well described experimental work, offering very useful practical information on the researched topic. It presents interesting results and findings, which can be useful to the researchers working on metal implants. This has publication merit, but a revision needs to be made to address the following minor issues:

English Language:

  • Many sentences are very long and become convoluted or difficult to follow. Please break down in smaller sentences and proof check for clarity.
  • Some expressions/terms are not correct (i.e. “…used worldwide”, instead of “used at worldwide level’ or ‘used globally). Please proofread.

Abstract:

  • It needs to be rewritten for clarity, for example:

- “To apply hot forging…’: This sentence needs to be rewritten (do not start a sentence with ‘To’.

- The second sentence is too long. Please rewrite

  1. Introduction
  • The paragraph contained between lines 54-70 needs to be adjusted or moved at this end of this section, as it refers not only to background (past studies) but also to work performed in this paper.
  1. Experimental Procedure
  • Section 2.1 (page 2): What do size 1, 4 etc refer to? Also, the stem length has to be shown in a drawing. Please provide figures and references for both.
  • Subsection 2.1.1: A table with the specimens’ produced, number, dimensions and roughness values (Ra) has to be added here.
  • Subsection 2.1.2: The laser power used with the EOS M290 and other processing parameters are missing. Please add.
  • Subsection 2.3.1: What TE stands for? Also, which test standard and how many specimens were used for the tensile tests? Please add both.
  • Subsection 2.3.2: Which test standard and how many specimens were used for the fatigue test? Please add.
  • Subsection 2.3.3: How many test coupons (hip stems) were used for the durability tests? Please add.
  • Subsection 2.4: This section has to be expanded to include more details on the immersion tests (including details on the standards and process used).
  • Section 2.5: The same applies here, more details are necessary (again, the number of specimens is not provided, which is a very important parameter for any statistical analysis).
  1. Results and Discussion
  • Section 3.1 has to be broken down to subsections describing each one the microstructural characteristics, tensile, fatigue and other properties examined separately for the various metals used (forged and AM), similarly to what you did in the experimental section (section 2).
  • Section 3.1: The first sentence (referring to Figure 1) seems unnecessary here. Please connect it in a better way with the rest of this section’s content or remove it.
  • Section 3.1: “L-N” curve? You refer to the P-N curve (as per vertical axis title of Figure 6). Please correct that.
  • Section 3.1: Are the deviation values (+/-495N, +/-512N and +/-463N) for the fatigue lives acceptable according to the standard used for this test? They seem to be quite significant when comparing with the mean values (3400, 6700 and 6400 N correspondingly). Please add further details/discussion about this finding.
  • Section 3.4: This should read ‘3.2’ (but subsections will be added anyway so it’s not too important)
  • Section 3.4: Lines 286-287 “Since the load F is inclined by 9° from the vertical direction…” Please add a figure showing this. It is not clear where one can see this angle.
  • Section 3.3: Firstly, this should be 3.5? Please correct. Also, in lines 338-339 “The mechanical properties of laser-sintered Ti-15-4-4 and Ti-6-4 rods are shown in Table 2(b) for comparison” the reader is referred back. This is confusing. Please check all sections for continuity and ease of reading.
  • Figure 12: What do we see here? What does the horizontal axis show? Also, it appears that there is no effect on the sintering process on the mechanical properties (the line is flat). Please elaborate and correct the caption / graph as necessary.
  • Many tables and figures run across two pages. Please correct (i.e. Table 3 in pages 17 and 18).
  1. Conclusions

No comments here – ok.

Author Response

Reviewer 2

Thank you for the peer review of the manuscript, which has been revised in accordance with your comments. The corrected parts are shown in red.

English Language

(1) We have reviewed the text and shortened the sentences.

(2) “worldwide” has been modified to “used globally”.

 Abstract: We have corrected it as pointed out by the reviewer (“To apply--- and second sentence”).

  1. Introduction: lines 54-70

We have removed these lines from Introduction and added then short to Section 2.1.1 after shortening the sentence.

  1. Experimental Procedure

 Section 2.1

 A new Table1 has been added showing the size. We have added an explanation about stem size.

 Section 2.1.1

 A new Table 1 has been added showing the size and Ra. We have added explanations about stem dimensions, the number of stems, and hot forging conditions.

Section 2.1.2

We have added typical selective laser melting conditions.

Subsection 2.3.1

We have added the test standard and the number of specimens for the tensile test.

Subsection 2.3.2

We have added the test standard and the number of specimens for the fatigue tests. Since the number can be found in Figures 7 and 12, we showed the minimum number of specimens. The number required for statistical processing is quite satisfactory.

Subsection 2.3.3

 We have added the test standard and the number of specimens for the durability tests. Since the number can be found in Figure 6, we showed the minimum number of specimens. The number required for statistical processing is quite satisfactory.

Subsection 2.4

We have added the details of the test standard and the method of the immersion test.

Subsection 2.5

We have added the test standard for statistical analysis that is similar as this analysis. The minimum number of specimens and the number of unbroken specimens are important factors. They are fully satisfied.

  1. Results and Discussion

Section 3

We have separated the hot forging and selective laser melting sections. The text composition, tables, and figures have been rearranged in the order of hot forging, selective laser melting, stress analysis, and immersion properties, as in the following sections. The related figures and tables have also been moved. The text has also been modified to make it easier to understand.

3.1. Microstructures and Mechanical Properties of Hot-Forged Artificial Hip Stems.

      3.1.1. Microstructures of Hot-Forged Artificial Hip Stems

      3.1.2. Mechanical Properties of Hot-Forged Artificial Hip Stems

3.2. Microstructures and Mechanical Properties of Selective-Laser-Melted Ti alloy Hip Stems and Rod  Specimens

      3.2.1. Microstructures of Selective-Laser-Melted Stems and Rod Specimens

      3.2.2. Mechanical Properties of Selective-Laser-Melted Stems and Rod Specimens

3.3. Stress Analysis of Artificial Hip Stem

3.4. Immersion Properties of Once-laser-melted Ti-15-4-4 and Wrought Ti-15-4 Plates

Section 3.1

The first sentence has been deleted.

Section 3.1

We have corrected it to the P-N curve.

Section 3.1

We have added a discussion on standard deviation.

Section 3.1

We have added new section names and modified section numbers to make the section content easier to read.

Section 3.4

The figure has been modified as a new Figure 15 to make it easier to understand the load points and tilt angles. It is shown. Figure 4 also shows a   diagram explaining the tilt angle.

Section 3.3

Stress analysis has been revised in new Section 3.3. Old Table 2 has been moved to the hot forging (new Table 3) and selective laser melting (new Table 4) sections.

Figure 12.

We have corrected the description of the horizontal axis in the figures and captions.

We have corrected the section name of section 3, positions of figures and tables in the manuscript to make the content easier to read.

Reviewer 3 Report

In this work, the microstructures, mechanical properties, fatigue strengths, and durability of hip stems in which specimens were annealed after hot forging or SLM have been investigated in order to apply hot forging and SLM to the manufacture of artificial hip joint stems with biocompatible Ti-15Zr-4Nb alloy (Ti-15-4). Design of experiment, data collection and data analysis have been carried out adequately. However, before further consideration, the following issues should be considered and addressed:

  1. Besides the properties, what would be the advantages of this development?
  2. It is highly recommended to use selective laser melting (SLM) instead of laser sintering. Because laser sintering makes confusion with the selective laser sintering (SLS) process that is a completely different process.
  3. Which method has been used for the chemical composition analysis? It should be described together with its sample preparation.
  4. If the chemical composition of the feedstock materials has been analyzed, it should be included and compared.
  5. The majority of the references are very old, and some recent works should be included in state of the art.
  6. The level of self-citation is very high (9/27)! The ratio should be lower than to less than 10-15%.

Author Response

Thank you for the peer review of the manuscript, which has been revised in accordance with your comments. The corrected parts are shown in red.

  1. There is a reduction in manufacturing cost. It is mentioned at the end of the Abstract and Introduction.
  2. The expression “laser sintering” has been modified to “selective laser melting (SLM)” through the text.
  3. We have added a description of the method of chemical analysis.
  4. We have added a description of feedstock materials for wrought materials to 2.1.1. The chemical compositions of the powder and laser-melted materials have been added to Table 2. The difference in chemical composition between the powder and laser-melted material has been described.
  5. We have searched for and cited new literature.
  6. We have minimized self-citation by citing other reports in the literature.

We have corrected the section name of section 3, positions of figures and tables in the manuscript to make the content easier to read.

Round 2

Reviewer 1 Report

The authors have revised the manuscript very well.

Author Response

Thank you for the peer review of our manuscript.

Reviewer 2 Report

This comment has not been addressed:

  • Section 3.1: Are the deviation values (+/-495N, +/-512N and +/-463N) for the fatigue lives acceptable according to the standard used for this test? They seem to be quite significant when comparing with the mean values (3400, 6700 and 6400 N correspondingly). Please add further details/discussion about this finding.

Author Response

Thank you for the peer review of the manuscript again. We have carefully revised it again. The corrections are as follows.

The standard deviation (SD) for the mean value of PD in this study was calculated using data of the entire P−N curve, and it is assumed that the SD is distributed to the same extent even for PD. The ratio of SD to the mean value of PD (SD/mean PD, 495/3400=0.15, 606/6800=0.09, 512/3000=0.17, 463/6400=0.07) was in the range of 7−17%. This SD/mean PD ratio for hip stems tended to be larger than the SD/mean σFS ratio shown in Table 3. This is considered to be due to the torsional force applied in addition to the compressive load in the durability test of the stems. As shown in Figure 16, the fatigue fracture from the edge may be related to these load and force. Further consideration of these factors may be needed in the future.

Reviewer 3 Report

The revision is satisfactory and thus the manuscript can be accepted in the current form.

Author Response

(The authors gave the same response as above.)
